# Chimeric crRNA improves CRISPR–Cas12a specificity in the N501Y mutation detection of Alpha, Beta, Gamma, and Mu variants of SARS-CoV-2

Jun Yang[1], Nilakshi Barua[1], Md. Nannur Rahman[1], Norman Lo[1], Tsz Fung Tsang[1], Xiao Yang[1], Paul K. S. Chan[1], Li Zhang[2], Margaret Ip[1]*

**1** Department of Microbiology, Faculty of Medicine, The Chinese University of Hong Kong, Prince of Wales Hospital, Hong Kong (SAR), China, **2** Department of Mechanical and Automation Engineering, The Chinese University of Hong Kong, Hong Kong (SAR), China

* margaretip@cuhk.edu.hk

**Data Availability Statement:** All relevant data are within the paper and its Supporting information files.

## Abstract

Many CRISPR/Cas platforms have been established for the detection of SARS-CoV-2. But the detection platform of the variants of SARS-CoV-2 is scarce because its specificity is very challenging to achieve for those with only one or a few nucleotide(s) differences. Here, we report for the first time that chimeric crRNA could be critical in enhancing the specificity of CRISPR-Cas12a detecting of N501Y, which is shared by Alpha, Beta, Gamma, and Mu variants of SARS-CoV-2 without compromising its sensitivity. This strategy could also be applied to detect other SARS-CoV-2 variants that differ only one or a few nucleotide(s) differences.

## Introduction

The emergence of SARS-CoV-2 variants, including B.1.1.7 (Alpha), B.1.351 (Beta), P.1 (Gamma), B.1.617.2 (Delta), C.37 (Lambda), and B.1.621 (Mu) continues to challenge infection control through accelerating transmission and/or escaping neutralization with the current vaccines [1–4]. It intensifies the urge to develop rapid detection tests that directly recognize these variants to triage those exposed for implementing measures of isolation or quarantine. CRISPR/Cas platform is an alternative and rapid method based on nucleic acid detection without expensive devices. It has been applied to SARS-CoV-2 and/or variants detection, including Cas9 (RAY [5]), Cas12a (DETECTR [6] and miSHELOCK [7]), and Cas13a (SHERLOCK [8]). Among them, the Cas12a system is faster than Cas13a as Cas13a requires *in vitro* transcription for detection, and possess higher target specificity than CRISPR-Cas9 system as the latter is more tolerant to mismatch [9]. Thus, CRISPR-Cas12a was used in our study. Detection of the N501Y plays a crucial role in identifying the Alpha, Beta, Gamma, and Mu variants of SARS-CoV-2 as N501Y mutation is shared by these variants. However, the hurdle for designing crRNA with high specificity for detecting N501Y is primarily due to single nucleotide substituting wild-type SARS-CoV-2. We report here a chimeric crRNA which could be efficiently used to detect N501Y with high specificity and sensitivity by CRISPR-Cas12a.

**Funding:** This work was supported by internal grants from the Department of Microbiology, Faculty of Medicine, The Chinese University of Hong Kong. The funders had no role in study design, data collection and analysis, decision to publish, or preparation of the manuscript.

**Competing interests:** The authors have declared that no competing interests exist.

## Results

We have designed and compared three different types of N501Y crRNA, namely, N501Y crRNA 20-nt (20-nt spacer, designing 20-nt spacer crRNAs is the traditional strategy for genome editing and detection by CRISPR-Cas12a, and in accordance, this 20-nt crRNA has recently been used to detect N501Y in miSHERLOCK platform [7]), N501Y chimeric crRNA 24-nt (24-nt spacer), and N501Y crRNA 24-nt (24-nt spacer) to determine the most efficient crRNA used for N501Y detection (The sequences of primers and crRNAs are given in Table 1). N501Y chimeric crRNA 24-nt was designed according to the method by Kim et al. replacing the last 8-nt of the crRNA with DNA which can improve CRISPR-Cas12a specificity of target DNA cleavage [9]. However, whether this kind of chimeric crRNA can increase the specificity of detection (collateral effect of Cas12a) has not been determined as per our knowledge. Therefore, N501Y crRNA 24-nt was also designed to evaluate whether the length of crRNA influences the specificity. We observed that the chimeric crRNA works best to differentiate N501Y from wild type as there is no false positive signal compared with a strong false positive signal for wild type when using N501Y crRNA 20-nt and N501Y crRNA 24-nt (Fig 1A). Thus, the chimeric crRNA enhances the specificity of the detection assay.

To determine whether the specificity and sensitivity of N501Y chimeric crRNA 24-nt is significant for N501Y detection, the limit of detection (LOD) was evaluated compared to N501Y crRNA 20-nt using the tenfold serial dilutions of synthetic RNA containing gene fragments of SARS-CoV-2. The fluorescent signal was evaluated by spectrophotometer (BioTeK Synergy H1 microplate reader, northern Vermont, USA) and the tubes were visualized by UV light of ChemiDoc™ Touch Imaging System (California, USA). Compared with N501Y crRNA 20-nt, the N501Y chimeric crRNA 24-nt achieved the same LOD (100 copies /μL RNA) for detecting N501Y samples (Fig 1B and 1D). Although the fluorescent signal of chimeric crRNA was lower than that of regular crRNA (Fig 1C and 1D), it could differentiate the signal between the N501Y and the wild type spectrophotometrically and visually without compromising the LOD. In the detection of samples containing wild type N501, N501Y crRNA 20-nt produced a weak false positive in the CRISPR-Cas12a reaction after incubation of 30 min (Fig 2A and 2B). After incubation for 1 h, the false positive signal is clearly visualized for $\geq 1 \times 10^3$ copies of RNA under UV light, as shown in Fig 1C. The fluorescent signal of the wild type was not visible when chimeric crRNA was used for tube detection under UV light (there was no false positive for chimeric crRNA even after 2 h incubation, Fig 2C and 2D). We observed similar

**Table 1. Sequences of primers and crRNAs used in this study.**

| crRNAs and primers | Sequences | Length (nt) |
|---|---|---|
| N501Y chimeric crRNA 24-nt | UAAUUUCUACUAAGUGUAGAUCAACCCACUUAUGGUG**TTGGTTAC** | 45 (The last 8-nt are DNA) |
| N501Y crRNA 24-nt | UAAUUUCUACUAAGUGUAGAUCAACCCACUUAUGGUGUUGGUUAC | 45 |
| N501Y crRNA 20-nt[a] | UAAUUUCUACUAAGUGUAGAUCAACCCACUUAUGGUGUUGG | 41 |
| N501Y RPA F | CAGGCCGGTAGCACACCTTGTAATGGTGTT | 30 |
| N501Y RPA R | TTGCTGGTGCATGTAGAAGTTCAAAAGAAAG | 31 |
| T7-3G IVT primer [10] | GAAATTAATACGACTCACTATAGGG | 25 |
| N501Y crRNA 24-nt IVT template | GTAACCAACACCATAAGTGGGTTGATCTACACTTAGTAGAAATTACCCTATAGTGAGTCGTATTAATTTC | 70 |
| N501Y crRNA 20-nt IVT template | CCAACACCATAAGTGGGTTGATCTACACTTAGTAGAAATTACCCTATAGTGAGTCGTATTAATTTC | 66 |

[a] The sequence is the same as N501Y crRNA of miSHERLOCK [7].

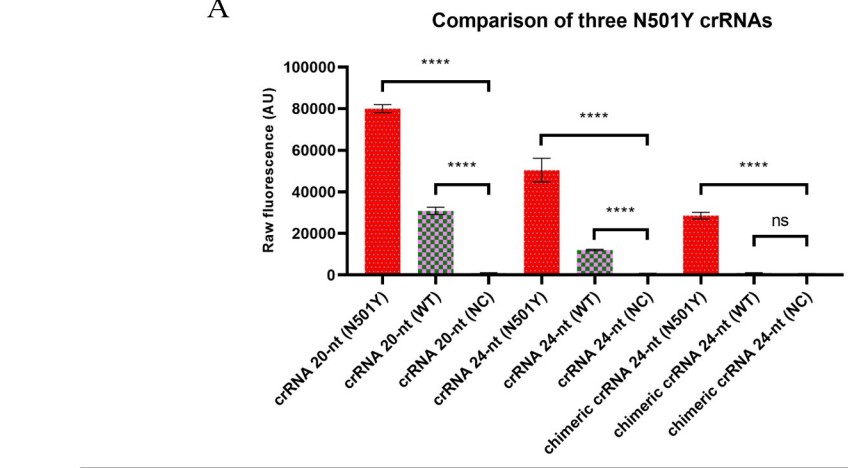

A  **Comparison of three N501Y crRNAs**

B

| Copy number/µL | 1E+05 | | 1E+04 | | 1E+03 | | 1E+02 | | 1E+01 | | 1E+0 | |
|---|---|---|---|---|---|---|---|---|---|---|---|---|
| crRNA type | crRNA | chimeric crRNA | crRNA | chimeric crRNA | crRNA | chimeric crRNA | crRNA | chimeric crRNA | crRNA | chimeric crRNA | crRNA | chimeric crRNA |
| N501Y (positive rate) | 12/12 | 12/12 | 12/12 | 12/12 | 12/12 | 12/12 | 12/12 | 12/12 | 4/12 | 4/12 | 0/12 | 0/12 |
| WT (positive rate) | 12/12 | 0/12 | 12/12 | 0/12 | 12/12 | 0/12 | 8/12 | 0/12 | 0/12 | 0/12 | 0/12 | 0/12 |

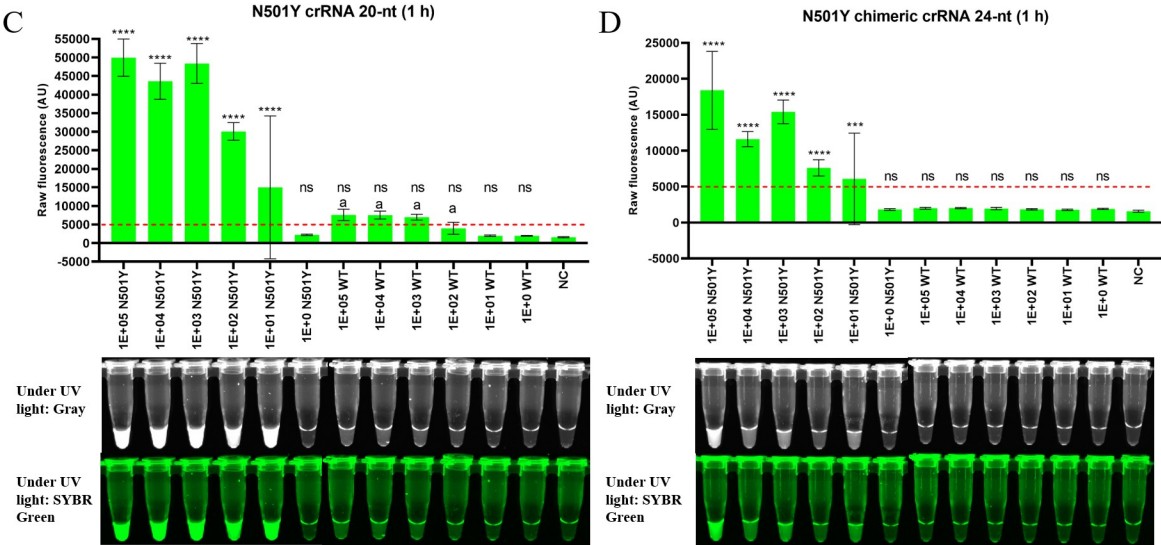

**Fig 1. Specificity and sensitivity of different crRNAs using synthetic RNA containing gene fragments of SARS-CoV-2.** (A) Detection of N501Y variant and wild type using N501Y chimeric crRNA 24-nt, crRNA 24-nt, and crRNA 20-nt after incubation at 37 ˚C for 10 min. Four replicates were run (n = 4). Recombinase Polymerase Amplification (RPA) using 1E+09 copies of synthetic DNA containing gene fragments of SARS-CoV-2 as a template. (B) Table summarizing positive rate (visualization of the tube) of N501Y and wild type when using different crRNAs after incubation for 1 h. (C) Detection of N501Y variant and wild type using N501Y crRNA 20-nt after incubation for 1 h. The (a) indicates false positive fluorescence signal of the wild type. (D) Detection of N501Y variant and wild type using N501Y chimeric crRNA 24-nt after incubation for 1 h. No false positive signal was obtained. The horizontal red dashed lines indicate the threshold fluorescence signal that can be visualized under UV light. The threshold level of 5000 AU has been determined for visualizing under UV light and validated against spectrophotometry reading. Tenfold serial dilutions (copy number per µL) of synthetic RNA containing gene fragments of SARS-CoV-2 was used for (B), (C), and (D). Four replicates were run for each time and repeated three times for (B), (C), and (D) (n = 12). NC stands for non-template control. Error bars represent the standard deviation of the mean. For (A), (C), and (D), statistical analysis was performed using a one-way ANOVA test with Dunnett's multiple comparisons test. Where the raw fluorescence (AU) of each reaction was compared to the respective NCs. The asterisks (*, **, ***, ****) indicate significant differences with $p < 0.05$, $p < 0.01$, $p < 0.001$, and $p < 0.0001$ and ns denotes not significant ($p > 0.05$).

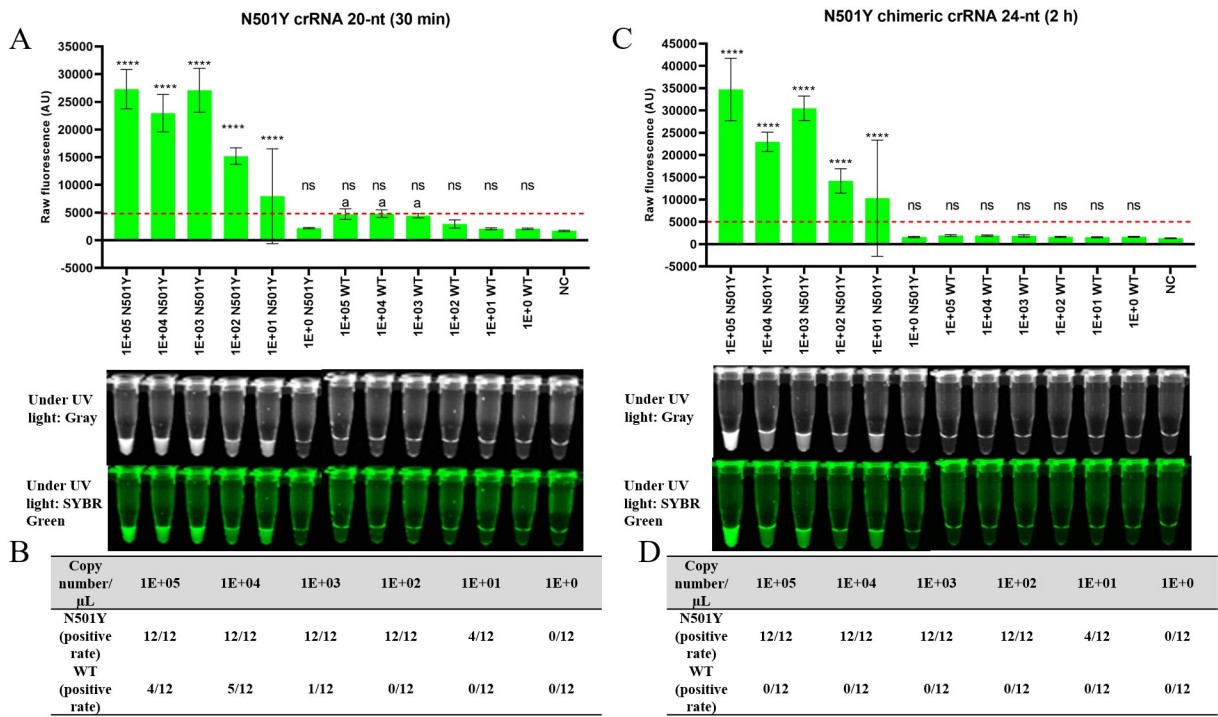

**Fig 2. Specificity of different crRNAs using synthetic RNA containing gene fragments of SARS-CoV-2.** (A) Detection of N501Y variant and wild type using N501Y crRNA 20-nt after incubation for 30 min. The (a) indicates false positive fluorescence signal of the wild type. (B) Table summarizing positive rate (visualization of the tube) of N501Y and wild type when using N501Y crRNA 20-nt after incubation for 30 min. (C) Detection of N501Y variant and wild type using N501Y chimeric crRNA 24-nt after incubation for 2 h. No false positive signal was obtained. (D) Table summarizing positive rate (visualization of the tube) of N501Y and wild type when using N501Y chimeric crRNA 24-nt after incubation for 2 h. The horizontal red dashed lines indicate the threshold fluorescence signal that can be visualized under UV light. The threshold level of 5000 AU has been determined for visualizing under UV light and validated against spectrophotometry reading. Tenfold serial dilutions (copy number per μL) of synthetic RNA containing gene fragments of SARS-CoV-2 was used. Four replicates were run for each time and repeated three times (n = 12). NC stands for non-template control. Error bars represent the standard deviation of the mean. For (A) and (C), statistical analysis was performed using a one-way ANOVA test with Dunnett's multiple comparisons test. Where the raw fluorescence (AU) of each reaction was compared to the respective NCs. The asterisks (*, **, ***, ****) indicate significant differences with $p < 0.05$, $p < 0.01$, $p < 0.001$, and $p < 0.0001$ and ns denotes not significant ($p > 0.05$).

results (LOD and false positive signal for N501Y crRNA 20-nt) using synthetic DNA containing gene fragments of SARS-CoV-2 (S1 Fig). The fluorescent signal could also be detected using blue LED light and orange acrylic goggles as a filter (S2 Fig). Thus, the chimeric crRNA can improve the specificity of CRISPR-Cas12a without compromising its sensitivity in detecting N501Y.

## Discussion

We report for the first time that chimeric crRNA can be used to efficiently differentiate N501Y of Alpha, Beta, Gamma, and Mu variants of SARS-CoV-2 from the wild type and other variants. The sensitivity of chimeric crRNA is comparable to regular crRNA, whereas the specificity is higher and can be stable for more than 2 h. It improves and provides the versatility of applying CRISPR-Cas12a in a larger throughput, even as a point-of-care testing (POCT). Replacing the last 8-nt of the crRNA with DNA can decrease the binding energy between crRNA and target DNA, leading to less off-target of CRISPR-Cas12a [9]. RNA-guided Cas12a unleashes indiscriminate single-stranded DNase activity when CRISPR-Cas12a recognizes its target [11]. As the detection signal is produced by cleavage of fluorophore-quenched ssDNA

fluorescent reporter, improving on-target specificity by chimeric crRNA 24-nt can increase the specificity of CRISPR-Cas12a detection. Designing chimeric crRNA can also be used for other variants detection with only one or a few nucleotide(s) differences. The primers and chimeric crRNA of our study can be developed into all-in-one tube detection using RPA and CRISPR-Cas12a. To further differentiate Alpha, Beta, Gamma, and Mu, other mutation sites in the Spike protein like 69–70 deletion or 144 deletion, 242–244 deletion or K417N, R190S or K417T, T95I or R346K can be detected respectively.

## Materials and methods

### Target RNAs and crRNAs preparation

Target RNAs were prepared from synthetic gene fragments of SARS-CoV-2 (Beijing Genomics Institute, BGI). The gene fragments were amplified using primer with T7 promoter by PCR followed by target RNAs synthesis through in vitro transcription (IVT, HiScribe T7 Quick High Yield RNA Synthesis Kit, NEB) using the PCR products. The IVT products were treated with TURBO DNase (Thermo Scientific) to remove the template and purified by Monarch RNA Cleanup Kit (NEB).

crRNAs were prepared by IVT (HiScribe T7 Quick High Yield RNA Synthesis Kit, NEB) using annealed crRNA templates with T7 promoter. The IVT products was purified as mentioned above for target RNAs.

### RPA and RT-RPA amplification

A 50 μL reaction mixture was prepared with 25 μL 2x Reaction Buffer (TwistAmp$^{®}$ Liquid Basic), 2.25 μL dNTPs (10 mM each dNTP, NEB, N0447S), 5 μL 10x Basic E-mix (TwistAmp$^{®}$ Liquid Basic), 2.4 μL each of the RPA primers (10 μM), 1.6 μL AMV Reverse Transcriptase (NEB: M0277L, only required for RT-RPA), 2.5 μL 20x Core Reaction Mix, 2.5 μL of 280mM MgOAc, and 5 μL RNA or DNA template, water to make up the volume to 50 μL. The reaction mix was incubated at 37 ˚C for 40 min and heated at 65 ˚C for 10 min.

### CRISPR-Cas12a detection

CRISPR-Cas12a detection mixture was prepared according to Broughton et al. [6]. In brief, 11.2 μL H2O, 2 μL 10 × NEB buffer 2.1, 0.8 μL EnGen Lba Cas12a (1 μM, NEB), 2 μL crRNA (400 nM) was premixed and incubated at 37 ˚C for 30 min. Then, 2 μL fluorophore-quenched ssDNA fluorescent reporter (1 μM, FAM-TTATTATT-BHQ1, BGI) and 2 μL of RPA products were added. Fluorescent intensities were monitored every 10 min at 37 ˚C for 2 h by BioTeK Synergy H1 microplate reader with 5'FAM channel (Excitation/Bandwidth: 484/12.5 and Emission/Bandwidth: 530/12.5). For tube detection, the tube was incubated at 37 ˚C before visualizing by ChemiDoc™ Touch Imaging System (Bio-Rad). The fluorescent signal of the tube was also visualized by blue LED light (with filter: an orange acrylic goggle (Invitrogen Safe Imager) was placed on top to filter out blue light and enhance contrast (S2 Fig). To enhance contrast a black paper with a non-reflection surface was used.

### Statistical analysis

Statistical analysis was performed using a one-way ANOVA test with Dunnett's multiple comparisons test by GraphPad Prism 9 (GraphPad Software, Inc.). A $p$ value $< 0.05$ was considered significant.

## Supporting information

**S1 Fig. Specificity and sensitivity of two type crRNAs using tenfold serial dilutions of synthetic DNA (copy number per µL) containing gene fragments of SARS-CoV-2.** (A) Detection of N501Y variant and wild type using N501Y crRNA 20-nt after incubation for 1 h. The (a) indicates fluorescence of the wild type could be visualized. (B) Detection of N501Y variant and wild type using N501Y chimeric crRNA 24-nt after incubation for 1 h. No false positive signal is obtained. The horizontal red dashed lines indicate the threshold fluorescence signal that can be visualized under UV light. The threshold level of 5000 AU has been determined for visualizing under UV light and validated against spectrophotometry reading. NC stands for non-template control. Error bars represent the standard deviation of means. (C) Table summarizing positive rate of N501Y and wild type when using different crRNAs after incubation for 1 h. Eight replicates were run (n = 8). For (A) and (B), statistical analysis was performed using a one-way ANOVA test with Dunnett's multiple comparisons test. Where the raw fluorescence (AU) of each reaction was compared to the respective NCs. The asterisks (\*, \*\*, \*\*\*, \*\*\*\*) indicate significant differences with $p < 0.05$, $p < 0.01$, $p < 0.001$, and $p < 0.0001$ and ns denotes not significant ($p > 0.05$).
(TIF)

**S2 Fig. Visualization of sample tubes by easily available materials.** (A) Equipment used for signal visualization, including blue LED light, filter (an orange acrylic goggles (Invitrogen Safe Imager)), sample tubes, and black paper with a non-reflective surface. (B) Photograph of P (positive) and N (negative) samples under blue LED light.
(TIF)

## Author Contributions

**Conceptualization:** Paul K. S. Chan, Li Zhang, Margaret Ip.

**Funding acquisition:** Margaret Ip.

**Investigation:** Jun Yang, Nilakshi Barua, Md. Nannur Rahman, Tsz Fung Tsang.

**Methodology:** Jun Yang.

**Project administration:** Margaret Ip.

**Resources:** Norman Lo, Xiao Yang, Paul K. S. Chan, Margaret Ip.

**Supervision:** Margaret Ip.

**Validation:** Nilakshi Barua, Md. Nannur Rahman.

**Visualization:** Jun Yang, Nilakshi Barua, Md. Nannur Rahman.

**Writing – original draft:** Jun Yang.

**Writing – review & editing:** Jun Yang, Nilakshi Barua, Md. Nannur Rahman, Margaret Ip.

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
