## [Decision Letter · Decision Letter 0]

8 Nov 2021

PONE-D-21-31939Chimeric crRNA improves CRISPR–Cas12a specificity in the N501Y mutation detection of Alpha, Beta, Gamma, and Mu variants of SARS-CoV-2PLOS ONE

Dear Dr. Ip,

Thank you for submitting your manuscript to PLOS ONE. After careful consideration, we feel that it has merit but does not fully meet PLOS ONE’s publication criteria as it currently stands. Therefore, we invite you to submit a revised version of the manuscript that addresses the points raised during the review process.

We look forward to receiving your revised manuscript.

Kind regards,

Yu-Hsuan Tsai

Academic Editor

PLOS ONE

Journal Requirements:

[This work was supported by internal grants from the Department of Microbiology, Faculty of Medicine, The Chinese University of Hong Kong.]

 [This work was supported by internal grants from the Department of Microbiology, Faculty of Medicine, The Chinese University of Hong Kong.]

Reviewers' comments:

Reviewer's Responses to Questions

**Comments to the Author**

1. Is the manuscript technically sound, and do the data support the conclusions?

Reviewer #1: Yes

Reviewer #2: Yes

2. Has the statistical analysis been performed appropriately and rigorously? 

Reviewer #1: No

Reviewer #2: Yes

3. Have the authors made all data underlying the findings in their manuscript fully available?

Reviewer #1: Yes

Reviewer #2: Yes

4. Is the manuscript presented in an intelligible fashion and written in standard English?

Reviewer #1: Yes

Reviewer #2: Yes

5. Review Comments to the Author

Reviewer #1: The baseline of the conclusion which chimeric crRNA improves the specificity of Cas12a-based detection of COVID-19 N501Y variant is acceptable and can be useful for the scientific community once the data is published. The following comments need to be addressed in the revision:

1. How does the LOD of 5,000 raw fluorescent units be selected? It seems that this is chosen in this paper with the intention to differentiate the detection result with crRNA 20-nt and chimeric crRNA 24-nt as false positive and negative, which makes the result more interesting. The more rigorous way to report specificity improvement using the fold change in background signal (e.g. ~ 3-fold) . Also, since the maximum signal of detection in chimeric crRNA also reduces, the author should compare the dynamic range of detection using different crRNA.

2. Statistical analysis of significance is not performed in all bar graphs where such analysis is needed and the method of statistical analysis is missing in the method section.

Reviewer #2: Overall comments & recommendations:

The aim of the study is the first report that chimeric crRNA could be useful for enhancing detection of the specificity of CRISPR-Cas12a for SARS-CoV-2 mutation N501Y, which is shared by Alpha, Beta, Gamma, and Mu variants of SARS-CoV-2 without reducing its sensitivity.

To test specificity, the authors developed N501Y chimeric crRNA 24-nt sequence and compared it with crRNA 20-nt to test N501Y variant and wild type of the virus. In addition, to determine whether the sensitivity of N501Y chimeric crRNA 24-nt is significant for N501Y detection, the limit of detection (LoD) was evaluated and compared to N501Y crRNA 20-nt using the tenfold serial dilution of synthetic RNA containing gene fragments of SARS-CoV-2.

The study provided valuable data on this chimeric crRNA. However, there are some comments on the study.

1.In the background section, the authors should explain how application of chimeric crRNA on N501Y detection is significant. And, the authors should further describe how to identify the variants of Alpha, Beta, Gamma, and Mu.

2. Please describe Statistical analysis in Method.

3.In Discussion, the effectiveness and mechanism of chimeric crRNA on N501Y is needed to discuss.

4.Lines149: Please describe the SARS-CoV-2 origins in the experiment

6. PLOS authors have the option to publish the peer review history of their article (what does this mean?). If published, this will include your full peer review and any attached files.

Reviewer #1: **Yes: **Zhuobin Liang

Reviewer #2: No

---

## [Author Response · Author response to Decision Letter 0]

16 Nov 2021

12th November, 2021

To Professor Emily Chenette 

Editor in Chief, 

PLOS ONE

Dear Prof. Emily Chenette,

Submission of revised manuscript entitled “Chimeric crRNA improves CRISPR–Cas12a specificity in the N501Y mutation detection of Alpha, Beta, Gamma, and Mu variants of SARS-CoV-2”

We enclose hereby the revised manuscript entitled “Chimeric crRNA improves CRISPR–Cas12a specificity in the N501Y mutation detection of Alpha, Beta, Gamma, and Mu variants of SARS-CoV-2” for considering to publish in your renowned journal PLOS ONE. 

We are very grateful for your comments and of the reviewers’. Our point-by-point response to the reviewers’ comments follow and our detailed revisions in the manuscript are highlighted with Tracked changes 

Journal Requirements:

Response: This manuscript has been prepared according to PLOS ONE’s style requirements and file naming format.

[This work was supported by internal grants from the Department of Microbiology, Faculty of Medicine, The Chinese University of Hong Kong.]

 [This work was supported by internal grants from the Department of Microbiology, Faculty of Medicine, The Chinese University of Hong Kong.]

Response: For your kind information, the funding-related text, “This work was supported by internal grants from the Department of Microbiology, Faculty of Medicine, The Chinese University of Hong Kong.” has been removed from the revised manuscript. The funders had no role in study design, data collection and analysis, decision to publish, or preparation of the manuscript.

Response: We do not wish to change our Data Availability statement.

Response: ‘Data not shown’ has been replaced by Fig 2 and we have added the data in the revised manuscript. Please refer to line 123 in the revised manuscript with track changes.

Response: The reference list is complete and correct. There is no retracted paper. We added Ref 10 and Ref 11 to the list. Please check the revised manuscript. For your kind information:

10. Kellner MJ, Koob JG, Gootenberg JS, Abudayyeh OO, Zhang F. SHERLOCK: nucleic acid detection with CRISPR nucleases. Nat Protoc. 2019;14: 2986–3012. doi:10.1038/s41596-019-0210-2

11. Chen JS, Ma E, Harrington LB, DaCosta M, Tian X, Palefsky JM, et al. CRISPR-Cas12a target binding unleashes indiscriminate single-stranded DNase activity. Science. 2018;360: 436–439. doi:10.1126/science.aar6245

Review Comments to the Author

Reviewer #1: The baseline of the conclusion which chimeric crRNA improves the specificity of Cas12a-based detection of COVID-19 N501Y variant is acceptable and can be useful for the scientific community once the data is published. The following comments need to be addressed in the revision:

1. How does the LOD of 5,000 raw fluorescent units be selected? It seems that this is chosen in this paper with the intention to differentiate the detection result with crRNA 20-nt and chimeric crRNA 24-nt as false positive and negative, which makes the result more interesting. The more rigorous way to report specificity improvement using the fold change in background signal (e.g. ~ 3-fold) . Also, since the maximum signal of detection in chimeric crRNA also reduces, the author should compare the dynamic range of detection using different crRNA.

Response: Many thanks for your comments. It is very important issue you raised here. We have explained in the manuscript how 5000 AU was chosen as the threshold. For your kind information, below 5000 AU from the spectrophotometry reading, these test tubes are not visible under UV light. The signal of the N501Y and wild type of chimeric crRNA 24-nt is lower than that of crRNA 20-nt, so the fold change cannot give the exact picture of specificity in this case. That is why we have provided the data with 5000 AU as the threshold. We did not mention any dynamic range because when the signal is below 5000 AU, it cannot be visualized whereas at any value above 5000 AU, it has been visualized.

2. Statistical analysis of significance is not performed in all bar graphs where such analysis is needed and the method of statistical analysis is missing in the method section.

Response: Thank you very much for your suggestions. Statistical analysis results have been added to all bar graphs. We also added the method of statistical analysis in the method section. Please check line 198 of the revised manuscript with track changes.

Reviewer #2: Overall comments & recommendations:

The aim of the study is the first report that chimeric crRNA could be useful for enhancing detection of the specificity of CRISPR-Cas12a for SARS-CoV-2 mutation N501Y, which is shared by Alpha, Beta, Gamma, and Mu variants of SARS-CoV-2 without reducing its sensitivity.

To test specificity, the authors developed N501Y chimeric crRNA 24-nt sequence and compared it with crRNA 20-nt to test N501Y variant and wild type of the virus. In addition, to determine whether the sensitivity of N501Y chimeric crRNA 24-nt is significant for N501Y detection, the limit of detection (LoD) was evaluated and compared to N501Y crRNA 20-nt using the tenfold serial dilution of synthetic RNA containing gene fragments of SARS-CoV-2.

The study provided valuable data on this chimeric crRNA. However, there are some comments on the study.

1.In the background section, the authors should explain how application of chimeric crRNA on N501Y detection is significant. And, the authors should further describe how to identify the variants of Alpha, Beta, Gamma, and Mu.

Response: Many thanks for your comments. As we are the first to use chimeric crRNA on detection in order to find a method that can improve the specificity of CRISPR-Cas12a detection, thus, we have added how the application of chimeric crRNA is significant in the discussion section. For your kind information: the sensitivity of chimeric crRNA is comparable to regular crRNA, whereas the specificity is higher and can be stable for more than 2 h. It improves and provides the versatility of applying CRISPR-Cas12a in a larger throughput which will take more time to prepare reactions, even as a point-of-care testing (POCT). Please refer to line 147 of the revised manuscript with track changes

How to identify the variants of Alpha, Beta, Gamma, and Mu has also been added in the discussion. It is worthy to mention that, N501Y mutation is shared by Alpha, Beta, Gamma, and Mu; identification of N501Y is the first step of differentiating variants of Alpha, Beta, Gamma, and Mu from wild type and other variants. To further differentiate Alpha, Beta, Gamma, and Mu, other mutation sites in the Spike protein like 69-70 deletion or 144 deletion, 242-244 deletion or K417N, R190S or K417T, T95I or R346K can be detected respectively. Please refer to line 158 of the revised manuscript with track changes

2. Please describe Statistical analysis in Method.

Response: Thank you very much for your valuable comments. We added the method of statistical analysis in the method section. Please check line 198 of the revised manuscript with track changes

3.In Discussion, the effectiveness and mechanism of chimeric crRNA on N501Y is needed to discuss.

Response: Many thanks for your comments. The effectiveness and mechanism of chimeric crRNA on N501Y have been added in the discussion of the revised manuscript. For your kind information: The sensitivity of chimeric crRNA is comparable to regular crRNA, whereas the specificity is higher and it is stable for more than 2 h. Replacing the last 8-nt of the crRNA with DNA can decrease the binding energy between crRNA and target DNA, leading to less off-target of CRISPR-Cas12a [9]. RNA-guided Cas12a unleashes indiscriminate single-stranded DNase activity when CRISPR-Cas12a recognizes its target [11]. As the detection signal is produced by cleavage of fluorophore-quenched ssDNA fluorescent reporter, improving on-target specificity by chimeric crRNA 24-nt can increase the specificity of CRISPR-Cas12a detection. Please refer to line 147 and line 150 of the revised manuscript with track changes.

4.Lines149: Please describe the SARS-CoV-2 origins in the experiment

Response: We have mentioned the SARS-CoV-2 origins in the revised manuscript (line 165). The above mentioned SARS-CoV-2 gene fragments was collected form BGI (Beijing Genomics Institute, a Chinese genome sequencing company, headquartered in Shenzhen, Guangdong, China). 

Yours sincerely,

Margaret Ip

Professor and Chairman,

Department of Microbiology,

The Chinese University of Hong Kong

Email: margaretip@cuhk.edu.hk

---

## [Decision Letter · Decision Letter 1]

7 Dec 2021

PONE-D-21-31939R1Chimeric crRNA improves CRISPR–Cas12a specificity in the N501Y mutation detection of Alpha, Beta, Gamma, and Mu variants of SARS-CoV-2PLOS ONE

Dear Dr. Ip,

Thank you for submitting your manuscript to PLOS ONE. After careful consideration, we feel that it has merit but does not fully meet PLOS ONE’s publication criteria as it currently stands. Therefore, we invite you to submit a revised version of the manuscript that addresses the points raised during the review process.

We look forward to receiving your revised manuscript.

Kind regards,

Yu-Hsuan Tsai

Academic Editor

PLOS ONE

Journal Requirements:

Reviewers' comments:

Reviewer's Responses to Questions

6. Review Comments to the Author

Reviewer #1: Minor suggestion: Figure 1C and 1D and the new Figure 2A and 2C look very similar, and my understanding is that they show results from the same experiments but at different time points after reaction incubation. To clarify the difference between Figure 1 and Figure 2 results, I recommend adding the time-point information directly on the figures.

---

## [Author Response · Author response to Decision Letter 1]

8 Dec 2021

Response: Many thanks for your suggestions. Time-point information has been added to Figure 1C and 1D and the new Figure 2A and 2C. Please check the revised Figure 1 and Figure 2. Time-point information has also been added to the legends of Figure 1 and S1 Figure. Please check lines 82 and 267 of the revised manuscript.

---

## [Editor Report · Decision Letter 2]

10 Dec 2021

Chimeric crRNA improves CRISPR–Cas12a specificity in the N501Y mutation detection of Alpha, Beta, Gamma, and Mu variants of SARS-CoV-2

PONE-D-21-31939R2

Dear Dr. Ip,

We’re pleased to inform you that your manuscript has been judged scientifically suitable for publication and will be formally accepted for publication once it meets all outstanding technical requirements.

Kind regards,

Yu-Hsuan Tsai

Academic Editor

PLOS ONE
---

## [Editor Report · Acceptance letter]

15 Dec 2021

PONE-D-21-31939R2 

Chimeric crRNA improves CRISPR–Cas12a specificity in the N501Y mutation detection of Alpha, Beta, Gamma, and Mu variants of SARS-CoV-2 

Dear Dr. Ip:

I'm pleased to inform you that your manuscript has been deemed suitable for publication in PLOS ONE. Congratulations! Your manuscript is now with our production department. 

Kind regards, 

on behalf of

Dr. Yu-Hsuan Tsai 

Academic Editor

PLOS ONE